# Amyloid and tau accumulate across distinct spatial networks and are differentially associated with brain connectivity

Joana B Pereira[1,2]*, Rik Ossenkoppele[2,3], Sebastian Palmqvist[2,4],
Tor Olof Strandberg[2], Ruben Smith[2,4], Eric Westman[1], Oskar Hansson[2,5]*

[1]Division of Clinical Geriatrics, Department of Neurobiology, Care Sciences and Society, Karolinska Institute, Stockholm, Sweden; [2]Clinical Memory Research Unit, Department of Clinical Sciences, Lund University, Malmö, Sweden; [3]Department of Neurology, Alzheimer Center, VU University Medical Center, Amsterdam, Netherlands; [4]Department of Neurology, Skåne University Hospital, Lund, Sweden; [5]Memory Clinic, Skåne University Hospital, Malmö, Sweden

**Abstract** The abnormal accumulation of amyloid-β and tau targets specific spatial networks in Alzheimer's disease. However, the relationship between these networks across different disease stages and their association with brain connectivity has not been explored. In this study, we applied a joint independent component analysis to [18]F- Flutemetamol (amyloid-β) and [18]F-Flortaucipir (tau) PET images to identify amyloid-β and tau networks across different stages of Alzheimer's disease. We then assessed whether these patterns were associated with resting-state functional networks and white matter tracts. Our analyses revealed nine patterns that were linked across tau and amyloid-β data. The amyloid-β and tau patterns showed a fair to moderate overlap with distinct functional networks but only tau was associated with white matter integrity loss and multiple cognitive functions. These findings show that amyloid-β and tau have different spatial affinities, which can be used to understand how they accumulate in the brain and potentially damage the brain's connections.

*For correspondence:
joana.pereira@ki.se (JBP);
oskar.hansson@med.lu.se (OH)

## Introduction

The accumulation of amyloid-β and tau deposits has been implicated in the genesis of Alzheimer's disease (*Jack et al., 2010*; *Jack et al., 2013*; *Jack et al., 2018*; *Jack and Holtzman, 2013*). These deposits appear and spread in the brain according to a characteristic spatial pattern: whereas amyloid-β deposits appear first in neocortical regions and spread to limbic and subcortical areas (*Thal et al., 2002*); tau deposits appear in the transentorhinal cortex and spread to paralimbic and neocortical areas (*Braak and Braak, 1991*). The relationship between these different and temporally dissociated spatial patterns is currently unclear but could provide an important insight into how they accumulate in the brain and contribute to cognitive decline in Alzheimer's disease.

Multimodal brain imaging offers a unique opportunity to explore the interrelation and overlap between distinct molecular pathologies and their functional consequences in vivo (*Drzezga et al., 2011*). Studies using positron emission tomography (PET) have shown that amyloid-β deposition occurs in highly connected brain areas, including the posterior cingulate, precuneus, anterior cingulate and medial orbitofrontal gyri (*Buckner et al., 2005*; *Buckner et al., 2009*; *Jagust and Mormino, 2011*; *Mormino et al., 2011*) already in asymptomatic stages (*Palmqvist et al., 2017*). In contrast, tau PET studies found that tau deposition occurs mainly in medial and inferior temporal areas, with

partial involvement of the parietal and occipital lobes in patients with mild cognitive impairment and Alzheimer's disease dementia (*Villemagne et al., 2015*; *Cho et al., 2016a*; *Johnson et al., 2016*; *Schöll et al., 2016*; *Mattsson et al., 2017*). These results suggest that amyloid-β and tau may target areas that belong to different functional networks (*Hansson et al., 2017*).

To our knowledge, the exact relationship between the spatial patterns of amyloid-β PET, tau PET and functional MRI within the same individuals is unknown, or between these patterns and white matter integrity. In addition, although a few studies have looked into the spatial patterns of amyloid-β and tau, none of them studied both of these patterns across the Alzheimer's disease continuum (*Jack et al., 2018*) in individuals that are cognitively normal as well as patients with mild cognitive impairment and dementia (*Myers et al., 2014*; *Brier et al., 2016*; *Jones et al., 2017*; *Sepulcre et al., 2017a*; *Hoenig et al., 2018*). This is important for several reasons, including the fact that amyloid-β and tau deposits accumulate in different brain regions with disease progression; thus, if different disease stages are not included, we might not be able to provide a complete overview of their spatial extent and topography or assess their full clinical value.

The aim of this study is to characterize the spatial networks associated with amyloid-β and tau accumulation across different stages of Alzheimer's disease. We applied a multivariate approach based on joint independent component analyses to identify linked amyloid-β and tau networks and assessed their relationship with resting-state functional MRI networks and white matter integrity in addition to cognition and gray matter atrophy. Based on previous evidence showing a strong link between amyloid-β pathology and functional connectivity (*Buckner et al., 2009*; *Jagust and Mormino, 2011*; *Myers et al., 2014*; *Elman et al., 2014*), we predicted there would be a greater spatial overlap between the amyloid-β and functional MRI networks compared to tau. Moreover, based on previously reported associations between tau pathology, white matter degeneration, gray matter atrophy and cognitive decline (*Villemagne et al., 2015*; *Bejanin et al., 2017*; *Aschenbrenner et al., 2018*; *Jacobs et al., 2018*; *Sintini et al., 2019*; *Strain et al., 2018*), we expected to find a stronger correlation between the tau networks, white matter integrity, gray matter volume and cognitive impairment.

## Results

One hundred seventeen individuals were included with $^{18}$F-Flutemetamol PET, $^{18}$F-Flortaucipir PET, structural T1-weighted MRI and neuropsychological data. This sample consisted of 26 cognitively normal subjects who were amyloid-β negative, in addition to 34 cognitively normal subjects, 21 patients with mild cognitive impairment and 36 patients with Alzheimer's disease dementia that were all amyloid-β positive (*Figure 1*). A subsample of this cohort also underwent resting-state functional MRI and diffusion tensor imaging (*Figure 1*). Amyloid-β positivity was established using a composite cortical region normalized by the whole cerebellum, brain stem and eroded subcortical white matter (*Landau et al., 2015*) on $^{18}$F-Flutemetamol PET images with a cut-off of >0.693.

The characteristics of the sample and differences between groups can be found in *Table 1*. As expected, there were no significant differences in cognition between amyloid-β negative and amyloid-β positive controls, but patients with mild cognitive impairment and Alzheimer's disease dementia showed worse scores in several cognitive tests.

### Amyloid-β and tau networks

The optimal number of components identified in the joint independent component analysis of $^{18}$F-Flutemetamol and $^{18}$F-Flortaucipir data was eleven using the minimum description length criterion (*Li et al., 2007*). This criterion selects the best hypothesis (a model and its parameters) for a given set of data as the one that leads to the best compression of the data. This method is provided within the Fusion ICA toolbox (*Calhoun et al., 2006*), which we used to perform the joint independent component analysis. The total variance explained by the amyloid-β components was 94.4% and the total variance explained by the tau components was 87.4%. Two of the independent components (IC 1, IC 9) were excluded from the analyses since they mainly included white matter areas for $^{18}$F-Flutemetamol data or off-target binding regions such as the basal ganglia for $^{18}$F-Flortaucipir data. The excluded components can be found in *Figure 2—figure supplement 1*. The remaining amyloid-β and tau components had major clusters in the gray matter and are described below.

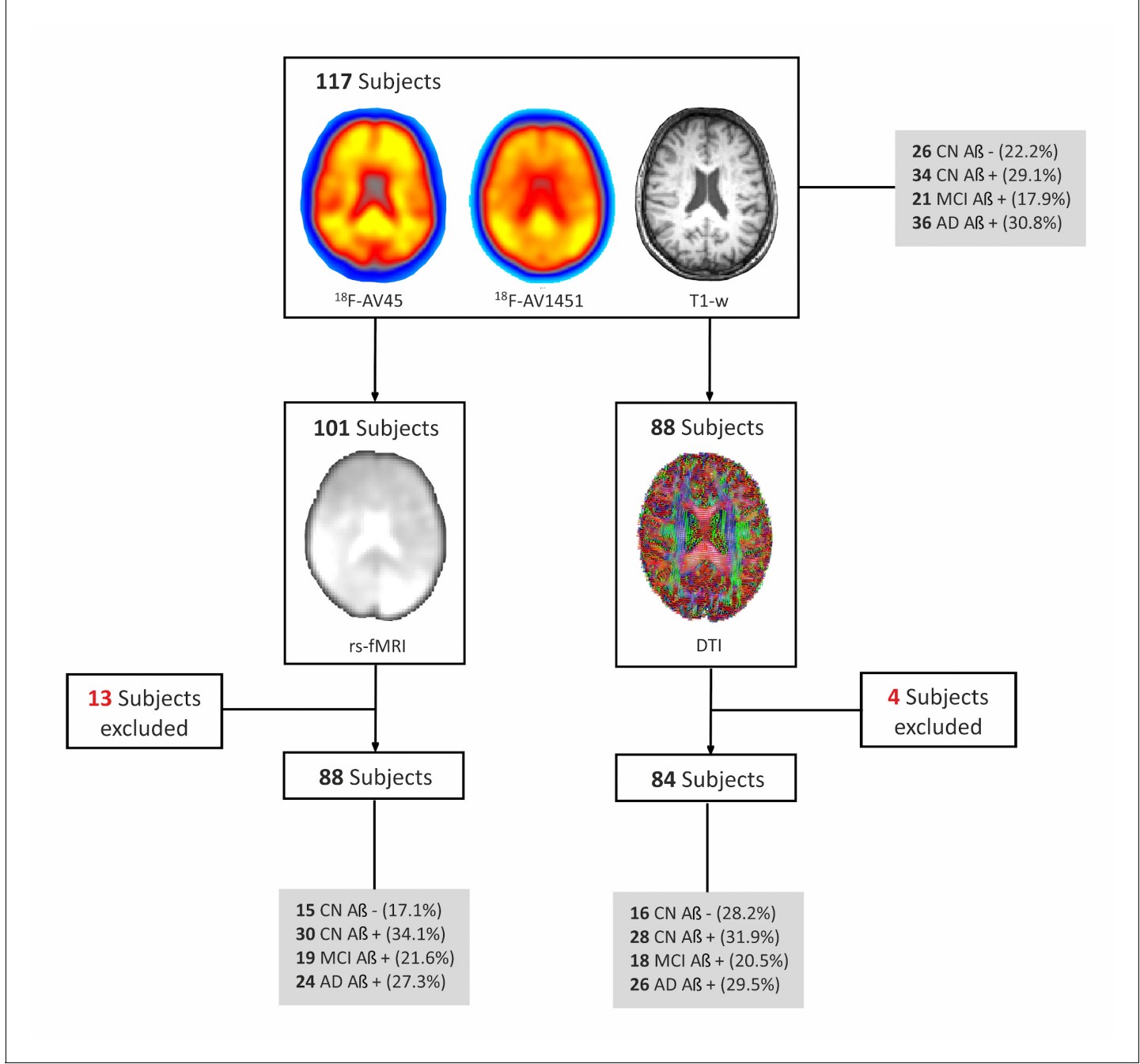

**Figure 1.** Flow-chart of imaging sequences available for the whole cohort. All subjects included in the study had a [18]F-Flutemetamol PET, [18]F-Flortaucipir ([18]F-AV1451) PET and T1-weighted (T1–w) MRI scans. In addition, a subsample (n = 101) also underwent resting-state functional MRI (rs-fMRI), of which 13 had to be excluded due to errors in spatial normalization or excessive head motion (see Materials and methods) so the final sample with rs-fMRI was 88 subjects. Finally, a subsample of subjects (n = 88) that had [18]F-Flutemetamol PET, [18]F-AV1451 PET and T1-w scans also underwent diffusion tensor imaging (DTI). Four subjects were excluded from the DTI subsample given they were outliers in white matter integrity measures. CN, cognitively normal; MCI, mild cognitive impairment; AD, Alzheimer's disease; amyloid negative (Aβ-); amyloid positive (Aβ+).

Three of the components showed similar spatial patterns for both amyloid-β and tau, and consisted of the left inferior occipital cortex (IC 2), the superior frontal gyrus (IC 5), and medial brain regions such as the precuneus and posterior cingulate (IC 6) (*Figure 2*). The other components included different areas for amyloid-β and tau, such as the medial fronto-parietal and temporal areas (IC 3), middle frontal and insular areas (IC 4), right parietal and fronto-parietal areas (IC 7), the

**Table 1.** Characteristics of the sample.

The values presented in the table correspond to means followed by (standard deviations). MMSE, mini-mental state examination. Amyloid positivity (Amyloid-β+) was established using a composite cortical region normalized by the whole cerebellum on $^{18}$F-Flutemetamol PET with a cut-off of >0.693. Subjects with values below this cut-off were classified as amyloid negative (Amyloid-β−). P values were calculated using Chi-square tests (to assess differences in sex) or Mann-Whitney tests (to assess differences in all other variables).

| | Cognitively normal (CN) | | Mild cognitive impairment (MCI) | Alzheimer's disease (AD) |
|---|---|---|---|---|
| | amyloid-β − (n = 26) | amyloid-β+ (n = 34) | All amyloid-β+ (n = 21) | All amyloid-β+ (n = 36) |
| Age (years) | 74.9 (5.7) | 75.2 (6.1) | 72.3 (6.9) | 70.6 (8.2)[b, d] |
| Sex (m/f) | 15/11 | 14/20 | 14/7 | 20/16 |
| Education (years) | 12.2 (3.5) | 11.7 (3.7) | 12.1 (3.5) | 12.2 (3.4) |
| MMSE | 28.8 (1.1) | 29.1 (1.1) | 26.0 (2.7)[a, c] | 20.6 (5.3)[b, d, e] |
| Delayed recall | 2.2 (1.7) | 2.6 (2.2) | 5.9 (2.4) [a,c] | 8.3 (2.0)[b, d, e] |
| Trail making test A | 44.4 (15.4) | 51.6 (19.5) | 59.4 (18.9)[a] | 83.4 (54.1)[b, d] |
| Clock-drawing test | 2.71 (1.19) | 2.80 (1.52) | 5.09 (1.08)[a, c] | 6.18 (1.58)[b, d] |

[a] Significant differences between amyloid negative controls and mild cognitive impairment patients (p<0.05).

[b] Significant differences between amyloid negative controls and Alzheimer's disease dementia patients (p<0.05).

[c] Significant differences between amyloid positive controls and mild cognitive impairment patients (p<0.05).

[d] Significant differences between amyloid positive controls and Alzheimer's disease dementia patients (p<0.05).

[e] Significant differences between mild cognitive impairment and Alzheimer's disease dementia patients (p<0.05).

anterior cingulate and hippocampus (IC 8), the posterior parietal and right inferior occipital areas (IC 9), and the lateral temporal and middle frontal areas (IC 11), respectively (*Figure 2*).

When the $^{18}$F-Flutemetamol SUVRs of each amyloid-β network were compared between groups, as expected, we found significant differences in all networks in the amyloid-β positive groups compared to amyloid-β negative controls (*Table 2*). Moreover, patients with mild cognitive impairment showed significantly higher $^{18}$F-Flutemetamol SUVRs in almost all networks compared to amyloid-β positive controls. Patients with Alzheimer's disease dementia showed significantly higher $^{18}$F-Flutemetamol SUVRs in all networks compared to amyloid-β positive controls, and in the right parietal (IC 7) and lateral temporal (IC 11) networks compared to patients with mild cognitive impairment (*Table 2*).

Interestingly, consistent with the earliest stages of tau deposition, we found higher $^{18}$F-Flortaucipir SUVRs in the temporal (IC 3) and hippocampal (IC 8) tau networks in amyloid-β positive compared to amyloid-β negative controls. Patients with mild cognitive impairment had significantly higher $^{18}$F-Flortaucipir SUVRs in almost all networks compared to controls that were amyloid-β negative or amyloid-β positive (*Table 2*). Patients with Alzheimer's disease had significantly higher $^{18}$F-Flortaucipir SUVRs in all networks compared to both amyloid-β negative and positive controls, and in almost all networks compared to patients with mild cognitive impairment (*Table 2*).

The SUVRs of the amyloid-β networks were positively correlated with the SUVRs of all tau networks (p<0.001).

## Overlap with functional MRI networks

The amyloid-β and tau patterns showed a fair to moderate overlap with the well-established functional networks provided by *Biswal et al. (2010)* (*Figure 3*). For instance, the amyloid-β medial fronto-parietal, medial, and posterior parietal networks overlapped moderately well with the posterior default-mode functional network (dice coefficients: 0.41, 0.24, 0.34), whereas the amyloid-β anterior cingulate network showed a fair overlap with the anterior cingulate functional network (dice coefficient: 0.30). The other amyloid-β networks showed a poor overlap with the rest of functional MRI networks, as can be observed in *Figure 3*.

In contrast to amyloid-β, the tau networks overlapped moderately well with a wider range of functional networks that did not include anterior or posterior parts of the default-mode network. These networks were the extra-striate visual network (dice coefficients: 0.33, 0.30), the primary visual

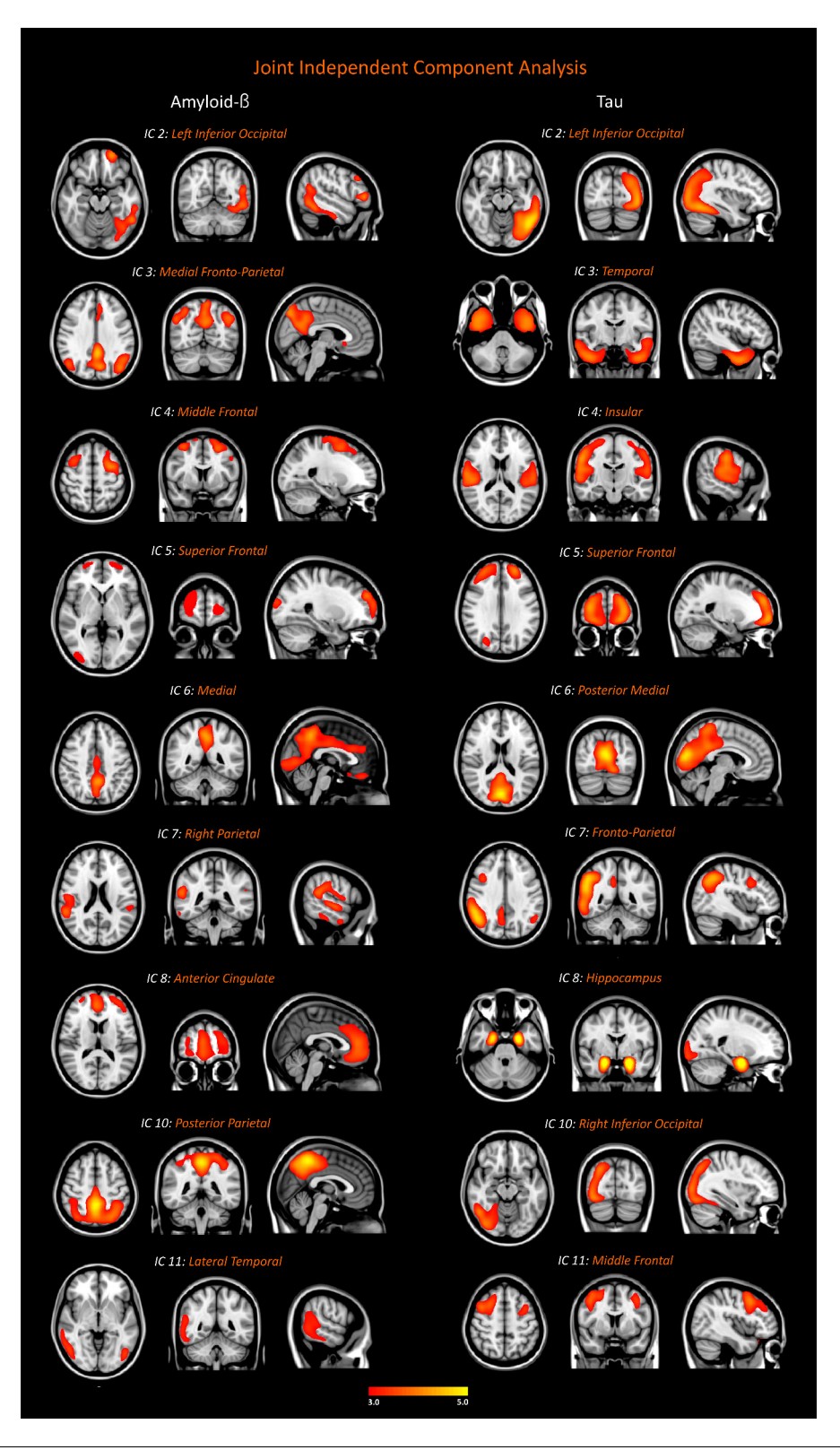

**Figure 2.** Networks of amyloid-β and tau accumulation in the whole cohort. We identified nine joint independent components (ICs) for amyloid-β and tau in the analyses of [18]F-Flutemetamol and [18]F-Flortaucipir PET data in the entire sample of 117 subjects (26 amyloid-β negative cognitively normal subjects, 34 amyloid-β positive cognitively normal subjects, 21 patients with mild cognitive impairment, 36 patients with Alzheimer's disease). These
*Figure 2 continued on next page*

*Figure 2 continued*

components were thresholded with a z-score >2.0, which corresponds to a two-tailed significance value of p<0.05. Inclusion and exclusion criteria for the subjects included in this analysis can be found in Materials and methods.

The online version of this article includes the following figure supplement(s) for figure 2:

**Figure supplement 1.** Excluded amyloid-β and tau components.
**Figure supplement 2.** Amyloid-β networks in different groups.
**Figure supplement 3.** Tau networks in different groups.
**Figure supplement 4.** Amyloid-β and tau networks in subsample with functional MRI and diffusion tensor imaging data.

network (dice coefficient: 0.41), the limbic network (dice coefficient: 0.39) the somatosensory network (dice coefficient: 0.48); the language network (dice coefficient: 0.37) and the right fronto-parietal network (dice coefficient: 0.24).

These findings indicate that the patterns of amyloid-β and tau accumulation show a moderate spatial affinity with distinct functional networks, with the amyloid-β patterns overlapping mainly with the default-mode network, whereas the tau patterns overlap fairly well with several different resting-state networks.

## Association with functional MRI network signals

To assess whether brain activation signals in our sample were associated with the amyloid-β and tau patterns, we extracted the time-series from the resting-state functional MRI images we had collected using as masks the functional networks from *Biswal et al. (2010)*. There were no significant correlations between the amyloid-β or tau SUVRs with the activation signals of the previous functional networks after adjusting for multiple comparisons. However, at an uncorrected level, we observed that a few amyloid-β networks such as the medial fronto-parietal, medial and posterior parietal networks correlated with the sensorimotor and dorsal attention functional networks, whereas the posterior medial and middle frontal tau networks correlated with the primary visual and extra-striate visual functional networks (*Supplementary file 1*).

Altogether, these results indicate there were very weak correlations between the PET network SUVRs and the functional network signals in our sample.

## Association with the integrity of white matter tracts

In contrast to the amyloid-β networks, which were not significantly associated with any white matter measures, the tau SUVRs correlated with fractional anisotropy in several white matter tracts. These tracts included the hippocampal cingulum tract, the inferior fronto-occipital fasciculus, the inferior longitudinal fasciculus, the superior longitudinal fasciculus, and the uncinate fasciculus (*Table 3*, *Figure 4*).

Moreover, we found several significant correlations between the tau network SUVRs and mean diffusivity in the cingulate cingulum tract, the hippocampal cingulum tract, the inferior fronto-occipital fasciculus, the inferior longitudinal fasciculus, and the superior longitudinal temporal fasciculus (*Table 3*, *Figure 5*).

## Relationship between the amyloid-β and tau networks with cognition

Our correlation analyses showed that worse global cognition (mini-mental state examination) was associated with increasing SUVRs in all tau networks and in almost all amyloid networks (*Table 4*), after adjusting for multiple comparisons. These associations were stronger for the right parietal amyloid-β network in addition to the left inferior occipital, temporal, fronto-parietal and hippocampus tau networks.

Regarding more specific cognitive functions, we found that worse episodic memory (delayed recall) correlated with increasing SUVRs in almost all amyloid-β and tau networks. As expected, the networks that showed a stronger association with memory were the temporal and hippocampus tau networks.

In contrast to amyloid-β, the tau networks also correlated with worse attention (trail making test A) and visuospatial (clock drawing test) abilities. We found that attention impairment was associated with greater tau SUVRs in the left and right inferior occipital, temporal, superior and middle frontal,

**Table 2.** Differences between groups in amyloid-β and tau networks.

The values presented in the table correspond to means followed by (standard deviations) or p values calculated using non-parametric permutation tests to assess differences between groups in amyloid-β and tau networks, while controlling for age and sex. Values in bold correspond to significant group differences after adjusting for multiple comparisons with false discovery rate corrections (FDR) (q < 0.05). The entire sample of 117 subjects were included in this analysis: 26 amyloid-β negative cognitively normal subjects, in addition to 34 amyloid-β positive cognitively normal subjects, 21 patients with mild cognitive impairment and 36 patients with Alzheimer's disease. Amyloid negative (Aβ-); amyloid positive (Aβ+). Amyloid positivity (β+) was established using a composite cortical region normalized by the whole cerebellum on $^{18}$F-Flutemetamol PET with a cut-off of >0.693.

| | Cognitively normal (CN) | | Mild cognitive impairment (MCI) | Alzheimer's disease (AD) | CN β− vs CN β+ | CN β− vs MCI | CN β− vs AD | CN β+ vs MCI | CN β+ vs AD | MCI vs AD |
|---|---|---|---|---|---|---|---|---|---|---|
| | amyloid-β− | amyloid-β+ | All amyloid-β+ | All amyloid-β+ | P value | P value | P value | P value | P value | P value |
| *amyloid-β networks* | | | | | | | | | | |
| IC 2: Left Inferior Occipital | 0.71 (0.04) | 0.92 (0.14) | 1.10 (0.10) | 1.13 (0.13) | **<0.001** | **<0.001** | **<0.001** | **<0.001** | **<0.001** | 0.287 |
| IC 3: Medial Fronto-Parietal | 0.67 (0.04) | 0.98 (0.16) | 1.15 (0.13) | 1.19 (0.14) | **<0.001** | **<0.001** | **<0.001** | **<0.001** | **<0.001** | 0.244 |
| IC 4: Middle Frontal | 0.74 (0.06) | 0.89 (0.12) | 0.94 (0.11) | 1.02 (0.16) | **<0.001** | **<0.001** | **<0.001** | 0.159 | **0.004** | 0.048 |
| IC 5: Superior Frontal | 0.68 (0.05) | 0.91 (0.14) | 1.02 (0.11) | 1.08 (0.16) | **<0.001** | **<0.001** | **<0.001** | **0.014** | **<0.001** | 0.101 |
| IC 6: Medial | 0.64 (0.04) | 0.92 (0.14) | 1.07 (0.13) | 1.13 (0.14) | **<0.001** | **<0.001** | **<0.001** | **<0.001** | **<0.001** | 0.112 |
| IC 7: Right Parietal | 0.69 (0.06) | 0.88 (0.14) | 1.00 (0.11) | 1.11 (0.12) | **<0.001** | **<0.001** | **<0.001** | **0.003** | **<0.001** | **0.001** |
| IC 8: Anterior Cingulate | 0.67 (0.05) | 0.99 (0.17) | 1.16 (0.14) | 1.23 (0.16) | **<0.001** | **<0.001** | **<0.001** | **0.003** | **<0.001** | 0.080 |
| IC 10: Posterior Parietal | 0.67 (0.05) | 0.95 (0.17) | 1.11 (0.16) | 1.15 (0.15) | **<0.001** | **<0.001** | **<0.001** | **0.002** | **<0.001** | 0.226 |
| IC 11: Lateral Temporal | 0.70 (0.04) | 0.95 (0.17) | 1.09 (0.14) | 1.18 (0.16) | **<0.001** | **<0.001** | **<0.001** | **0.006** | **<0.001** | **0.033** |
| *tau networks* | | | | | | | | | | |
| IC 2: Left Inferior Occipital | 1.15 (0.07) | 1.21 (0.10) | 1.76 (0.62) | 2.03 (0.58) | 0.112 | **<0.001** | **<0.001** | **<0.001** | **<0.001** | 0.061 |
| IC 3: Temporal | 1.15 (0.06) | 1.22 (0.13) | 1.65 (0.42) | 1.83 (0.34) | **0.020** | **<0.001** | **<0.001** | **<0.001** | **<0.001** | 0.057 |
| IC 4: Insular | 1.00 (0.07) | 1.02 (0.07) | 1.17 (0.20) | 1.39 (0.48) | 0.151 | **0.030** | **<0.001** | 0.115 | **<0.001** | **0.010** |
| IC 5: Superior Frontal | 1.11 (0.08) | 1.14 (0.07) | 1.32 (0.34) | 1.58 (0.48) | 0.267 | **0.038** | **<0.001** | 0.060 | **<0.001** | **0.014** |
| IC 6: Posterior Medial | 1.05 (0.06) | 1.09 (0.07) | 1.30 (0.34) | 1.57 (0.57) | 0.157 | **0.019** | **<0.001** | 0.066 | **<0.001** | **0.015** |
| IC 7 Fronto-Parietal | 1.13 (0.07) | 1.20 (0.13) | 1.63 (0.60) | 1.97 (0.60) | 0.498 | **0.001** | **<0.001** | **0.002** | **<0.001** | **0.020** |
| IC 8: Hippo-campus | 1.11 (0.06) | 1.18 (0.12) | 1.46 (0.30) | 1.67 (0.39) | **0.018** | **<0.001** | **<0.001** | **0.002** | **<0.001** | **0.012** |
| IC 10: Right Inferior Occipital | 1.12 (0.07) | 1.16 (0.08) | 1.57 (0.57) | 1.91 (0.68) | 0.143 | **0.001** | **<0.001** | **0.004** | **<0.001** | **0.034** |
| IC 11: Middle Frontal | 1.07 (0.07) | 1.10 (0.08) | 1.39 (0.34) | 1.71 (0.51) | 0.188 | **0.001** | **<0.001** | **0.003** | **<0.001** | **0.003** |

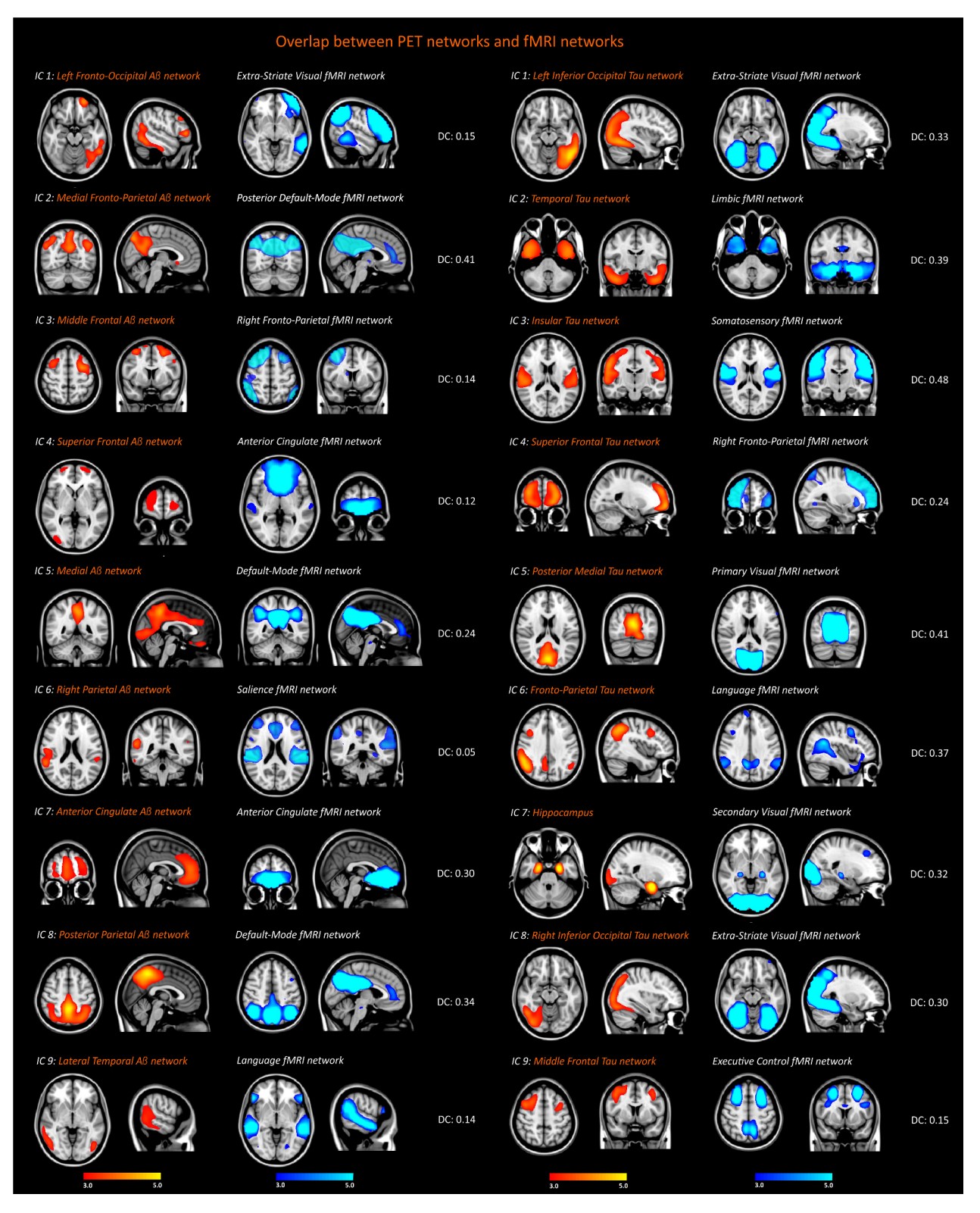

**Figure 3.** Spatial overlap between the functional MRI, amyloid-β and tau networks. Resting-state functional networks that overlapped best with the amyloid-β and tau networks identified in the entire sample of 117 subjects (26 amyloid-β negative cognitively normal subjects, 34 amyloid-β cognitively normal subjects, 21 patients with mild cognitive impairment, 36 patients with Alzheimer's disease). The Dice coefficients (DC) between these networks indicated poor (<0.2), fair (0.2–0.4) or moderate (0.4–0.6) overlap.

**Table 3.** Association between the integrity of white matter tracts and the PET networks.

The values presented in the table correspond to Spearman's Rho followed by (p values) for the correlations between the fractional anisotropy or mean diffusivity of white matter tracts and PET networks SUVRs, while controlling for age, sex and presence of cognitive impairment. Values in bold correspond to significant group differences after adjusting for multiple comparisons with false discovery rate corrections (FDR) (q < 0.05). Correlations were carried out in all amyloid-β positive individuals: 28 amyloid-β positive cognitively normal subjects (CN Aβ+), 18 patients with mild cognitive impairment (MCI Aβ+) and 26 patients with Alzheimer's disease (AD Aβ+). Three AD patients and 1 MCI patient were excluded due to being outliers in white matter measures. ATR, anterior thalamic radiation; CST, cortico-spinal tract; CCT, cingulate cingulum tract; HCT, hippocampal cingulum tract; FMAJ, forceps major; FMIN, forceps minor; IFO, inferior fronto-occipital fasciculus; ILF, inferior longitudinal fasciculus; SLF, superior longitudinal fasciculus; SLFt, temporal part of the superior longitudinal fasciculus; UNC, uncinate fasciculus.

| | Fractional anisotropy | | | | | | | | | | |
| | ATR | CST | CCT | HCT | FMAJ | FMIN | IFO | ILF | SLF | SLFt | UNC |
|---|---|---|---|---|---|---|---|---|---|---|---|
| amyloid-β IC 2 | −0.016 (0.904) | 0.081 (0.527) | 0.014 (0.915) | 0.031 (0.810) | 0.011 (0.930) | 0.079 (0.540) | −0.093 (0.467) | −0.073 (0.569) | −0.111 (0.386) | −0.123 (0.336) | −0.047 (0.717) |
| amyloid-β IC 3 | −0.162 (0.205) | 0.133 (0.298) | 0.027 (0.832) | −0.052 (0.685) | −0.094 (0.462) | −0.091 (0.478) | −0.180 (0.159) | −0.105 (0.411) | −0.158 (0.217) | −0.211 (0.097) | −0.073 (0.572) |
| amyloid-β IC 4 | −0.121 (0.347) | 0.026 (0.841) | −0.043 (0.740) | −0.107 (0.404) | −0.112 (0.384) | −0.110 (0.393) | −0.166 (0.195) | −0.076 (0.552) | −0.130 (0.310) | −0.187 (0.142) | −0.042 (0.743) |
| amyloid-β IC 5 | −0.025 (0.848) | 0.089 (0.488) | 0.006 (0.964) | −0.027 (0.836) | −0.073 (0.568) | 0.018 (0.890) | −0.085 (0.507) | −0.002 (0.989) | −0.036 (0.781) | −0.125 (0.331) | 0.014 (0.911) |
| amyloid-β IC 6 | −0.177 (0.166) | 0.085 (0.509) | 0.033 (0.799) | −0.009 (0.947) | −0.123 (0.338) | −0.072 (0.575) | −0.164 (0.199) | −0.080 (0.533) | −0.123 (0.337) | −0.194 (0.128) | −0.061 (0.636) |
| amyloid-β IC 7 | 0.007 (0.957) | 0.004 (0.978) | −0.071 (0.582) | −0.046 (0.719) | −0.062 (0.630) | 0.022 (0.863) | −0.072 (0.576) | −0.031 (0.813) | −0.018 (0.890) | −0.064 (0.617) | −0.051 (0.692) |
| amyloid-β IC 8 | −0.017 (0.896) | 0.065 (0.611) | 0.024 (0.852) | 0.045 (0.725) | −0.003 (0.979) | 0.052 (0.687) | −0.037 (0.774) | 0.024 (0.851) | −0.003 (0.981) | −0.073 (0.572) | 0.029 (0.824) |
| amyloid-β IC 10 | −0.170 (0.183) | 0.082 (0.525) | 0.038 (0.765) | −0.062 (0.627) | −0.152 (0.234) | −0.072 (0.576) | −0.175 (0.170) | −0.082 (0.524) | −0.140 (0.275) | −0.175 (0.170) | −0.039 (0.760) |
| amyloid-β IC 11 | 0.040 (0.757) | 0.064 (0.618) | −0.040 (0.760) | −0.045 (0.724) | −0.020 (0.874) | 0.096 (0.456) | −0.083 (0.517) | −0.087 (0.497) | −0.070 (0.585) | −0.093 (0.467) | −0.046 (0.718) |
| tau IC 2 | −0.214 (0.093) | −0.123 (0.336) | −0.291 (0.021) | **−0.364 (0.003)** | −0.283 (0.025) | −0.170 (0.184) | −0.307 (0.014) | **−0.429 (<0.001)** | −0.339 (0.007) | −0.259 (0.041) | −0.280 (0.026) |
| tau IC 3 | −0.207 (0.104) | −0.092 (0.476) | −0.286 (0.023) | **−0.398 (0.001)** | −0.251 (0.047) | −0.234 (0.065) | −0.301 (0.017) | **−0.362 (0.004)** | −0.257 (0.042) | −0.283 (0.025) | −0.292 (0.020) |
| tau IC 4 | −0.222 (0.081) | −0.305 (0.015) | −0.309 (0.014) | −0.348 (0.005) | −0.292 (0.020) | −0.165 (0.198) | −0.339 (0.007) | **−0.459 (<0.001)** | **−0.378 (0.002)** | −0.297 (0.018) | −0.303 (0.016) |
| tau IC 5 | −0.266 (0.035) | −0.245 (0.053) | −0.370 (0.003) | **−0.361 (0.004)** | −0.271 (0.032) | −0.288 (0.022) | **−0.395 (0.001)** | **−0.430 (<0.001)** | **−0.386 (0.002)** | −0.286 (0.023) | **−0.398 (0.001)** |
| tau IC 6 | −0.224 (0.077) | −0.176 (0.167) | −0.239 (0.059) | −0.309 (0.014) | −0.281 (0.026) | −0.150 (0.241) | −0.292 (0.020) | **−0.393 (0.001)** | −0.330 (0.008) | −0.214 (0.093) | −0.225 (0.077) |
| tau IC 7 | −0.226 (0.075) | −0.194 (0.127) | −0.352 (0.005) | **−0.424 (<0.001)** | −0.337 (0.007) | −0.232 (0.068) | **−0.375 (0.003)** | **−0.448 (<0.001)** | **−0.370 (0.003)** | −0.272 (0.031) | −0.324 (0.010) |
| tau IC 8 | −0.064 (0.619) | −0.094 (0.464) | −0.245 (0.053) | −0.323 (0.010) | −0.234 (0.065) | −0.049 (0.702) | −0.222 (0.080) | −0.319 (0.011) | −0.215 (0.091) | −0.171 (0.180) | −0.179 (0.160) |
| tau IC 10 | −0.258 (0.041) | −0.121 (0.344) | −0.212 (0.096) | −0.304 (0.016) | −0.324 (0.010) | −0.136 (0.287) | −0.311 (0.013) | **−0.427 (<0.001)** | −0.350 (0.005) | −0.250 (0.048) | −0.223 (0.081) |

*Table 3 continued on next page*

*Table 3 continued*

**Fractional anisotropy**

| | ATR | CST | CCT | HCT | FMAJ | FMIN | IFO | ILF | SLF | SLFt | UNC |
|---|---|---|---|---|---|---|---|---|---|---|---|
| tau IC 11 | −0.166 (0.192) | −0.188 (0.140) | −0.300 (0.017) | −0.306 (0.015) | −0.208 (0.102) | −0.145 (0.257) | −0.310 (0.013) | **−0.392 (0.002)** | −0.347 (0.005) | −0.271 (0.032) | −0.299 (0.017) |

**Mean Diffusivity**

| | ATR | CST | CCT | HCT | FMAJ | FMIN | IFO | ILF | SLF | SLFt | UNC |
|---|---|---|---|---|---|---|---|---|---|---|---|
| amyloid-β IC 2 | 0.031 (0.808) | −0.156 (0.223) | −0.061 (0.636) | 0.011 (0.931) | −0.132 (0.302) | 0.048 (0.712) | −0.072 (0.573) | 0.011 (0.930) | −0.042 (0.742) | 0.087 (0.500) | 0.078 (0.543) |
| amyloid-β IC 3 | 0.053 (0.679) | −0.362 (0.004) | −0.154 (0.227) | 0.020 (0.879) | −0.216 (0.089) | −0.051 (0.691) | −0.108 (0.398) | −0.028 (0.829) | −0.168 (0.188) | 0.082 (0.525) | 0.065 (0.613) |
| amyloid-β IC 4 | 0.067 (0.601) | −0.143 (0.263) | −0.029 (0.824) | 0.092 (0.476) | 0.043 (0.739) | −0.055 (0.669) | 0.039 (0.763) | 0.088 (0.495) | 0.011 (0.935) | 0.101 (0.433) | 0.034 (0.793) |
| amyloid-β IC 5 | 0.029 (0.820) | −0.120 (0.348) | 0.001 (0.995) | 0.014 (0.915) | 0.003 (0.983) | −0.029 (0.823) | −0.018 (0.886) | 0.024 (0.850) | −0.057 (0.656) | 0.059 (0.645) | 0.029 (0.823) |
| amyloid-β IC 6 | 0.128 (0.318) | −0.278 (0.028) | −0.085 (0.508) | 0.088 (0.493) | −0.125 (0.331) | 0.021 (0.869) | −0.016 (0.899) | 0.050 (0.698) | −0.083 (0.519) | 0.148 (0.249) | 0.122 (0.342) |
| amyloid-β IC 7 | 0.081 (0.530) | −0.088 (0.493) | 0.085 (0.507) | 0.096 (0.456) | −0.057 (0.655) | 0.108 (0.401) | 0.066 (0.605) | 0.141 (0.272) | 0.106 (0.409) | 0.108 (0.398) | 0.118 (0.358) |
| amyloid-β IC 8 | 0.011 (0.932) | −0.115 (0.368) | −0.046 (0.718) | −0.005 (0.972) | 0.037 (0.771) | −0.009 (0.943) | −0.038 (0.767) | 0.025 (0.847) | −0.085 (0.507) | 0.009 (0.944) | 0.015 (0.907) |
| amyloid-β IC 10 | 0.088 (0.494) | −0.310 (0.013) | −0.112 (0.381) | 0.057 (0.657) | −0.211 (0.097) | −0.056 (0.661) | −0.068 (0.595) | −0.025 (0.844) | −0.142 (0.266) | 0.107 (0.403) | 0.063 (0.625) |
| amyloid-β IC 11 | 0.049 (0.703) | −0.096 (0.453) | 0.034 (0.791) | 0.023 (0.858) | −0.238 (0.060) | 0.125 (0.329) | −0.053 (0.679) | 0.037 (0.771) | 0.068 (0.598) | 0.102 (0.428) | 0.134 (0.296) |
| tau IC 2 | 0.266 (0.035) | 0.039 (0.760) | 0.324 (0.010) | 0.353 (0.005) | −0.011 (0.930) | 0.244 (0.054) | 0.271 (0.032) | 0.307 (0.015) | 0.183 (0.152) | 0.432 (<0.001) | 0.303 (0.016) |
| tau IC 3 | 0.219 (0.085) | 0.016 (0.900) | 0.289 (0.022) | 0.354 (0.004) | 0.041 (0.748) | 0.080 (0.532) | 0.250 (0.048) | 0.331 (0.008) | 0.171 (0.181) | 0.351 (0.005) | 0.183 (0.150) |
| tau IC 4 | 0.172 (0.177) | 0.054 (0.673) | 0.280 (0.026) | 0.222 (0.081) | 0.016 (0.899) | 0.164 (0.201) | 0.236 (0.062) | 0.243 (0.055) | 0.093 (0.471) | 0.303 (0.016) | 0.259 (0.040) |
| tau IC 5 | 0.228 (0.072) | 0.195 (0.126) | **0.387 (0.002)** | 0.260 (0.039) | 0.072 (0.576) | 0.186 (0.145) | 0.362 (0.004) | 0.348 (0.005) | 0.262 (0.038) | **0.436 (<0.001)** | 0.337 (0.007) |
| tau IC 6 | 0.226 (0.075) | −0.006 (0.960) | 0.261 (0.039) | 0.265 (0.036) | −0.074 (0.565) | 0.210 (0.099) | 0.242 (0.056) | 0.245 (0.054) | 0.116 (0.365) | **0.382 (0.002)** | 0.298 (0.018) |
| tau IC 7 | 0.261 (0.039) | 0.083 (0.519) | **0.418 (<0.001)** | **0.371 (0.003)** | 0.086 (0.501) | 0.243 (0.055) | **0.371 (0.003)** | **0.403 (0.001)** | 0.288 (0.022) | **0.463 (<0.001)** | 0.319 (0.011) |
| tau IC8 | 0.107 (0.405) | 0.084 (0.514) | 0.311 (0.013) | 0.253 (0.046) | 0.025 (0.848) | 0.148 (0.248) | 0.202 (0.112) | 0.232 (0.067) | 0.158 (0.216) | 0.281 (0.026) | 0.174 (0.172) |
| tau IC 10 | 0.252 (0.046) | −0.054 (0.672) | 0.302 (0.016) | 0.327 (0.009) | −0.052 (0.685) | 0.212 (0.096) | 0.225 (0.076) | 0.239 (0.059) | 0.074 (0.563) | **0.388 (0.002)** | 0.285 (0.024) |
| tau IC 11 | 0.193 (0.129) | 0.056 (0.663) | 0.314 (0.012) | 0.258 (0.042) | −0.075 (0.557) | 0.259 (0.040) | 0.226 (0.075) | 0.299 (0.017) | 0.199 (0.118) | **0.431 (<0.001)** | 0.350 (0.005) |

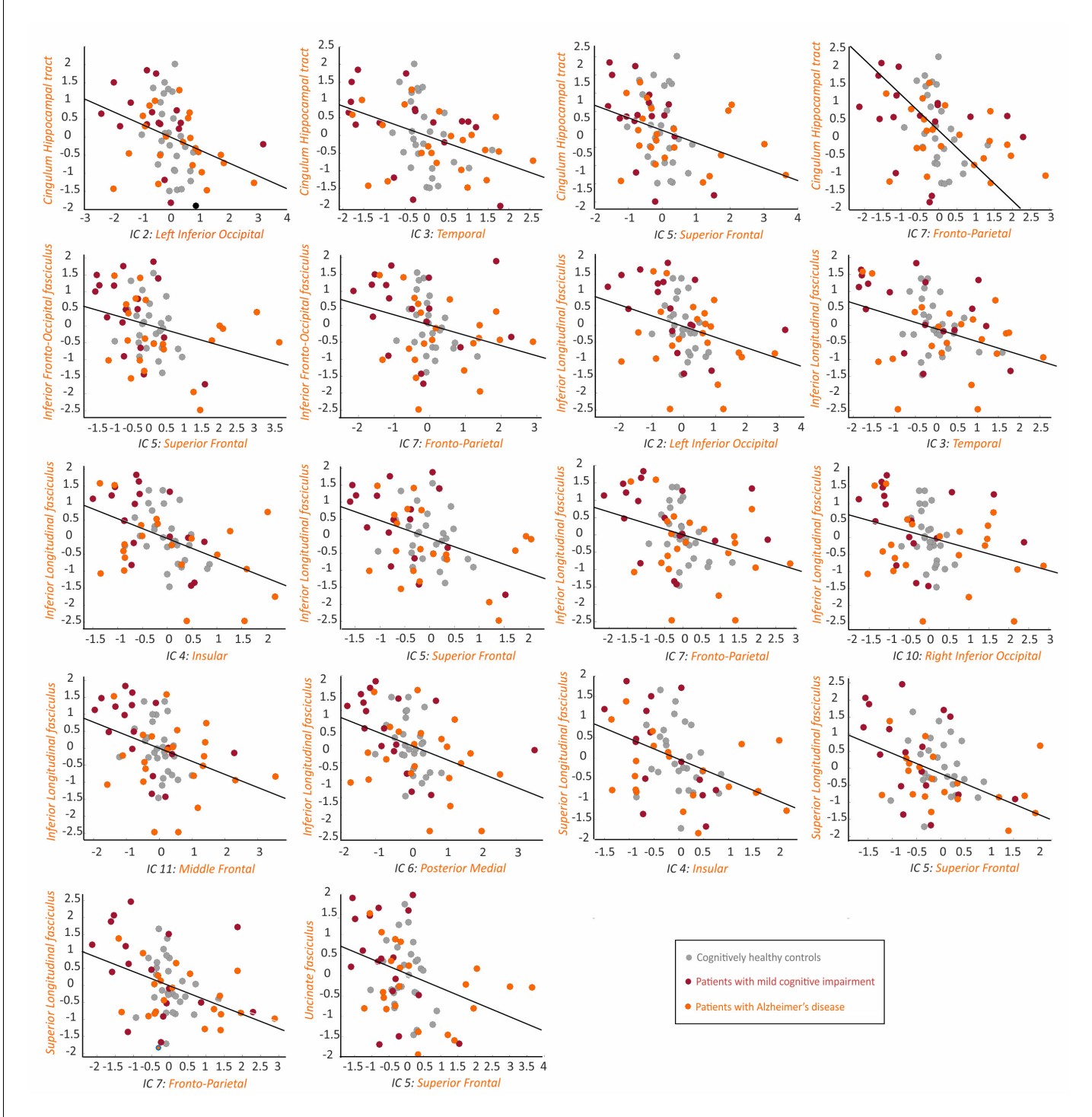

**Figure 4.** Association between the tau networks and fractional anisotropy of white matter tracts. The plots show significant correlations between the SUVRs of different tau networks and fractional anisotropy values of white matter tracts in 66 amyloid-β positive subjects with both imaging modalities, after regressing out the effects of age, gender and cognitive impairment, and adjusting for multiple comparisons using FDR corrections (q < 0.05). Three outliers were excluded: 2 AD patients and 1 MCI patient.

The online version of this article includes the following source data for figure 4:

**Source data 1.** Association between the tau networks and fractional anisotropy of white matter tracts.

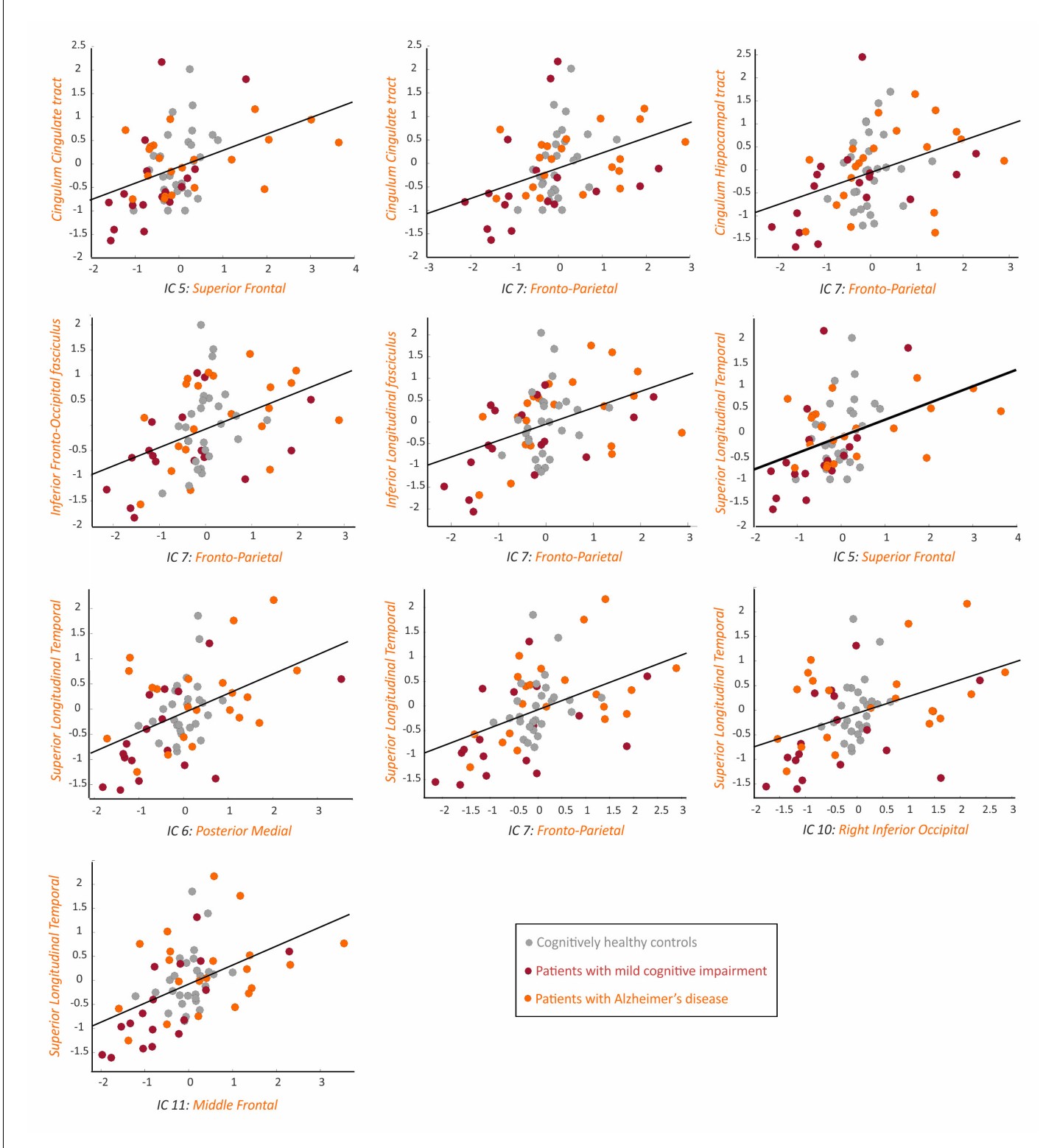

**Figure 5.** Association between the tau networks and mean diffusivity of white matter tracts. The plots show significant correlations between the SUVRs of different tau networks and mean diffusivity values of white matter tracts in 66 amyloid-β positive subjects with both imaging modalities, after regressing out the effects of age, gender and cognitive impairment, and adjusting for multiple comparisons using FDR corrections (q < 0.05). One AD patient with high mean diffusivity values in white matter tracts was excluded from all plots.

The online version of this article includes the following source data for figure 5:

**Source data 1.** Association between the tau networks and mean diffusivity of white matter tracts.

**Table 4.** Association between cognition with the amyloid-β and tau networks.

The values presented in the table correspond to Spearman's Rho followed by (p values) for correlations between cognitive tests and PET networks SUVRs, while controlling for age, sex, education and presence of cognitive impairment. Correlations were carried out across all amyloid-β positive individuals: 34 amyloid-β positive cognitively normal subjects (CN Aβ+), 21 patients with mild cognitive impairment (MCI Aβ+) and 36 patients with Alzheimer's disease (AD Aβ+). Some patients had missing scores on the MMSE (3 AD Aβ+), Delayed memory test (1 MCI Aβ+, 5 AD Aβ+), Trail-Making test (1 CN Aβ−, 3 MCI Aβ+, 5 AD Aβ+) and the Clock-drawing test (1 CN Aβ−, 10 CN Aβ+, 11 MCI Aβ+, 11 AD Aβ+). Values in bold correspond to significant group differences after adjusting for multiple comparisons with false discovery rate corrections (FDR) (q < 0.05).

| | MMSE Rho (p value) | Delayed recall Rho (p value) | Trail making test Rho (p value) | Clock-drawing test Rho (p value) |
|---|---|---|---|---|
| amyloid-β IC 2: Left Inferior Occipital | **−0.269 (0.013)** | **0.245 (0.028)** | −0.076 (0.508) | −0.040 (0.773) |
| amyloid-β IC 3: Medial Fronto-Parietal | −0.175 (0.111) | **0.261 (0.019)** | −0.062 (0.590) | 0.006 (0.965) |
| amyloid-β IC 4: Middle Frontal | −0.103 (0.351) | 0.179 (0.113) | −0.003 (0.690) | 0.023 (0.870) |
| amyloid-β IC 5: Superior Frontal | **−0.292 (0.007)** | **0.280 (0.012)** | 0.005 (0.978) | −0.055 (0.689) |
| amyloid-β IC 6: Medial | **−0.396 (<0.001)** | **0.284 (0.011)** | −0.006 (0.958) | −0.093 (0.498) |
| amyloid-β IC 7: Right Parietal | **−0.292 (0.007)** | **0.284 (0.011)** | −0.054 (0.639) | −0.111 (0.420) |
| amyloid-β IC 8: Anterior Cingulate | −0.166 (0.131) | **0.325 (0.003)** | −0.119 (0.301) | −0.038 (0.783) |
| amyloid-β IC 10: Posterior Parietal | **−0.308 (0.004)** | 0.191 (<0.089) | 0.032 (0.779) | −0.069 (0.617) |
| amyloid-β IC 11: Lateral Temporal | **−0.393 (<0.001)** | 0.214 (0.057) | 0.037 (0.751) | −0.042 (0.759) |
| tau IC 2: Left Inferior Occipital | **−0.413(<0.001)** | **0.338 (0.002)** | **0.311 (0.006)** | **−0.406 (0.002)** |
| tau IC 3: Temporal | **−0.246 (0.005)** | **0.446 (<0.001)** | **0.277 (0.014)** | **−0.271 (0.046)** |
| tau IC 4: Insular | **−0.274 (0.024)** | 0.169 (0.134) | 0.206 (0.070) | **−0.322 (0.017)** |
| tau IC 5: Superior Frontal | **−0.286 (0.008)** | **0.268 (0.017)** | **0.325 (0.004)** | −0.176 (0.199) |
| tau IC 6: Posterior Medial | **−0.325 (0.003)** | **0.305 (0.006)** | **0.285 (0.011)** | **−0.302 (0.025)** |
| tau IC 7: Fronto-Parietal | **−0.382 (<0.001)** | **0.344 (0.002)** | **0.417 (<0.001)** | **−0.321 (0.017)** |
| tau IC 8: Hippocampus | **−0.414 (<0.001)** | **0.439 (<0.001)** | 0.225 (0.047) | −0.229 (0.093) |
| tau IC 10: Right Inferior Occipital | **−0.339 (0.002)** | **0.279 (0.012)** | **0.315 (0.005)** | **−0.338 (0.012)** |
| tau IC 11: Middle Frontal | **−0.415 (<0.001)** | **0.344 (0.002)** | **0.265 (0.019)** | **−0.311 (<0.021)** |

medial and fronto-parietal networks. Moreover, we found that visuospatial impairment correlated with increasing tau SUVRs in the left occipital, insular, medial, fronto-parietal and middle frontal networks (*Table 4*).

For the amyloid and tau networks that were significantly associated with cognition (global cognition, delayed recall), we applied Fisher's tests to determine the difference between the pairs of significant correlation coefficients. These analyses showed there were no significant differences between the correlations of amyloid and tau with global cognition and delayed recall, suggesting that the effects of both pathologies were not dissociable from each other in these specific correlations. These findings are in line with previous evidence showing that amyloid-β and tau interact with each other to increase brain damage (*Ittner and Götz, 2011*), and ultimately cognitive dysfunction.

## Association with gray matter volumes in regions vulnerable to Alzheimer's disease

There were significant correlations between gray matter volumes in regions that are vulnerable to Alzheimer's disease (*Darby et al., 2019*), and the SUVRs of all tau networks (*Supplementary file 2*). Moreover, there was also a significant correlation between gray matter volumes and the SUVRs of the amyloid-β right parietal network (*Supplementary file 2*). These findings agree with previous evidence showing that, compared to amyloid-β, tau pathology is more closely association with neuronal loss or neurodegeneration (*Gómez-Isla et al., 1997*; *Giannakopoulos et al., 2003*; *Bejanin et al., 2017*).

## Discussion

The interplay between amyloid-β and tau has been extensively studied in Alzheimer's disease (*Ittner and Götz, 2011*). There is clear evidence that these proteins start accumulating several years before the appearance of clinical symptoms and spread between brain areas with disease progression (*Braak and Braak, 1998*; *Thal et al., 2002*; *Hyman et al., 2012*; *Jack et al., 2014*; *Braak and Del Tredici, 2015*; *Jansen et al., 2015*). In line with this, neuropathological evidence has shown that amyloid-β and tau have different spatial affinities and target distinct brain regions (*Braak and Braak, 1991*). In this study, we identified these different spatial patterns in vivo and assessed how they relate to functional and anatomical brain connectivity. Our results showed that amyloid-β and tau have distinct accumulation patterns but also overlap in a few brain regions. Moreover, both of them showed a fair to moderate overlap with distinct functional MRI networks, but only the tau patterns were associated with white matter integrity. These findings indicate that amyloid-β and tau deposits not only have distinct spatial patterns but their accumulation might have a different impact on functional and anatomical brain connectivity.

The recent development of tau PET tracers has provided us the unique opportunity to combine them with amyloid-β PET tracers and assess their specific contributions to Alzheimer's disease pathogenesis. Most of the studies carried out so far have focused on the accumulation of amyloid-β and tau PET in independent voxels or brain regions (*Cho et al., 2016a*; *Cho et al., 2016b*; *Johnson et al., 2016*; *Schöll et al., 2016*; *Schöll et al., 2017*; *Ossenkoppele et al., 2016*; *Wang et al., 2016*; *Mattsson et al., 2017*; *Mattsson et al., 2018*), despite the fact that their accumulation in these voxels and regions are presumably not independent. In the current study, we were able to capture these dependencies in amyloid-β and tau by applying an unsupervised method based on joint independent component analysis. As a result, we identified a number of linked amyloid-β and tau networks that resembled, to a large extent, the neuropathological stages describing the spread of plaques and tangles in Alzheimer's disease (*Braak and Braak, 1991*). For instance, for amyloid-β we identified a medial, anterior cingulate, posterior parietal and medial fronto-parietal networks that matched early amyloid stages (*Palmqvist et al., 2017*) in addition to five neocortical networks that matched more advanced amyloid stages (*Thal et al., 2002*). Regarding tau, we identified temporal, hippocampal, insular and neocortical networks, which agreed well with the spread of tangles from medial temporal to limbic and neocortical regions (*Braak and Braak, 1991*; *Hyman et al., 2012*). These findings suggest that our amyloid-β and tau networks reflected the accumulation of pathological changes occurring in postmortem studies in patients at early and advanced stages of Alzheimer's disease. This was further confirmed by our group analyses, which showed significant differences in amyloid-β and tau burden not only between patients and controls (which has already been shown by previous studies: *Cho et al., 2016a*; *Johnson et al., 2016*; *Schöll et al., 2016*; *Wang et al., 2016*), but also between amyloid-β positive and negative controls, and between patients with Alzheimer's disease and patients with mild cognitive impairment. Moreover, the associations we found between the amyloid-β and tau networks with cognition indicate that they also have an important value in characterizing clinical impairment. For amyloid-β, there were significant correlations between global cognition and memory decline, whereas for tau there were additional correlations with executive and visuospatial impairment. Of note, we did not find any significant associations between cognition and global amyloid-β PET, one of the most used markers of amyloid pathology in Alzheimer's disease. This provides further support to the use of amyloid-β networks like the ones identified in this study since they offer greater sensitivity to clinical impairment compared to conventional amyloid measures.

Amyloid-β and tau pathologies have been consistently described as accumulating across interconnected brain networks in Alzheimer's disease (*Seeley et al., 2009*; *Zhou et al., 2012*). A number of studies using functional MRI have supported this by showing that the regions of the default-mode network could be important hubs through which the disease progresses in the brain (*Buckner et al., 2009*; *Hedden et al., 2009*; *Sperling et al., 2009*; *Sheline et al., 2010*; *Drzezga et al., 2011*; *Mormino et al., 2011*; *Sheline and Raichle, 2013*). Our findings indicate that the amyloid-β patterns overlapped moderately well with the anterior and posterior hubs of the default-mode network, in line with previous studies showing that the brain regions showing highest amyloid-β accumulation coincide with the regions of this functional network like the precuneus and posterior cingulate (*Buckner et al., 2005*; *Buckner et al., 2009*; *Palmqvist et al., 2017*). In contrast, the tau patterns

overlapped fairly or moderately well with a wider range of functional networks such as the visual, limbic, somatosensory, language and fronto-parietal networks. These findings suggest that tau could have a higher spatial affinity with several functional networks, which could potentially explain why it correlates with impairment in more cognitive domains in contrast to amyloid-β. Thus, as previously suggested (*Blennow and Hampel, 2003*), tau could be a better stage marker that reflects the progression of Alzheimer's disease through different brain regions and clinical stages.

Interestingly, in this study we found that the accumulation of amyloid-β and tau converged in three brain areas such as the left inferior occipital cortex, superior frontal gyri and the precuneus. The focus on amyloid-β and tau has shifted from studying their separate toxic roles to understanding their possible interactions. There are different studies showing that amyloid-β exacerbates the toxic effects of tau and that tau can also mediate amyloid-β toxicity (*Ittner and Götz, 2011*). Here, we suggest that the precuneus, superior frontal and left inferior occipital gyri might be vulnerable to the combined toxic effects of amyloid-β and tau since they were present in both amyloid-β and tau network analyses. In a previous study, it was also found that amyloid-β and tau converge in a few brain regions such as the precuneus (*Sepulcre et al., 2016*), in line with our findings.

Another interesting finding of our joint independent component analyses is that the temporal tau network was linked to the medial fronto-parietal amyloid-β network, which in turn overlapped moderately well with the resting-state default-mode network. Since the accumulation of tau in temporal regions and alterations of the default-mode network are both important imaging markers that can be observed since early stages of Alzheimer's disease (*Sperling et al., 2009*; *Sepulcre et al., 2017a*; *Sepulcre et al., 2017b*), it is interesting to see that they both came out as connected components in our analyses, which could potentially reflect the interaction between the two proteinopathies in early disease stages. Future studies using longitudinal PET imaging are needed to further explore this finding and assess whether the link between these components changes over time or predicts clinical progression.

Our correlation analyses with gray matter volumes and white matter integrity suggest that, compared to amyloid-β, tau accumulation in Alzheimer's disease is more closely associated with brain atrophy and white matter degeneration. Several studies have shown that tau pathology increases both locally and distally to brain regions with synaptic connections (*Ahmed et al., 2014*; *Khan et al., 2014*; *Hu et al., 2016*), and is responsible for neuronal loss (*Gómez-Isla et al., 1997*; *Giannakopoulos et al., 2003*) and axonal degeneration (*de Calignon et al., 2012*). Our findings are in line with previous studies since we found that different tau networks were associated with lower fractional anisotropy and higher mean diffusivity in several white matter tracts, some of which were spatially close to the tau patterns (such as the temporal tau pattern and the cingulum hippocampal tract), whereas others were distant (such as the fronto-parietal tau pattern and the inferior occipito-frontal fasciculus).

We would like to highlight that a large part of the analyses carried out in the current study were based on multiple correlation analyses in all amyloid-β positive individuals. By performing these analyses, we aimed to evaluate the strength of the relationship between different imaging and clinical markers across the Alzheimer's disease continuum. According to the amyloid cascade hypothesis proposed by *Jack et al. (2013)*, individuals that are cognitively normal progress to mild cognitive impairment and Alzheimer's disease due to increasing amyloid accumulation, tau accumulation and MRI changes. In addition, according to the recent NIA-AA criteria (*Jack et al., 2018*), AD should be regarded as a continuum rather than three distinct clinically defined entities (cognitively normal, mild cognitive impairment, dementia). By carrying out correlation analyses across all amyloid-β positive subjects, our goal was to obtain information about how different PET, MRI and clinical markers behave with disease progression, in line with the recent NIA-AA guidelines for AD (*Jack et al., 2018*). However, it should be noted that these analyses do not provide information about causality or temporal ordering between variables. Thus, our analyses do not allow drawing definitive conclusions regarding the spread of amyloid-β and tau across different brain networks.

Moreover, there are a few other issues that should be recognized in our study. First of all, although we acquired different imaging sequences from the same individuals, the functional MRI and diffusion tensor imaging scans were only available for a subsample of subjects in our study. Our secondary analyses (*Figure 2—figure supplement 4*) showed that the PET networks identified in this subsample were similar to the ones in the whole cohort but it would still have been better to have the same number of subjects for all imaging modalities. Secondly, our study is cross-sectional

so our findings need to be replicated in longitudinal studies to assess whether the amyloid-β and tau networks can predict connectivity changes and cognitive decline over time. The weak correlations we found between amyloid-β and tau SUVRs with the signals of functional MRI networks was unexpected and suggests that perhaps our network analyses are not well suited to study these type of associations.

Despite these limitations, in this study we provide key evidence that amyloid-β and tau accumulate across distinct spatial networks that are clinically meaningful and closely associated with cognitive decline. In addition, we provide important clues on their potential overlap with different functional networks and their association with anatomical brain connectivity as well as gray matter atrophy. In line with the ongoing debate on the underlying mechanisms of Alzheimer's disease (*Seeley et al., 2009*; *Zhou et al., 2012*), our findings shed some light on which networks are preferentially targeted by amyloid-β and tau as well as their clinical value in distinguishing different disease stages and predicting cognitive decline.

## Materials and methods

### Participants

The subjects included in the current study were part of the prospective and longitudinal Swedish BioFINDER study (http://biofinder.se/), which was designed to identify and develop new markers for neurodegenerative disorders, particularly Alzheimer's disease. In addition to amyloid-β imaging using $^{18}$F- Flutemetamol PET, all subjects underwent tau imaging using $^{18}$F-Flortaucipir PET and T1-weighted MRI, and a substantial part underwent resting-state functional MRI and diffusion tensor imaging. All individuals were selected using the criteria described below.

Cognitively normal subjects were required to have a clinical dementia rating score of 0, 27–30 points on the mini-mental state examination, not fulfill criteria for mild cognitive impairment or dementia, have no history of cognitive change over time, and be fluent in Swedish (*Palmqvist et al., 2017*).

Only patients with mild cognitive impairment that was thought to be caused by Alzheimer's disease were included. These patients fulfilled the DSM-5 criteria for mild neurocognitive disorder and possible Alzheimer's disease. In addition, they were required to show abnormal amyloid-β accumulation on $^{18}$F-Flutemetamol PET or cerebrospinal fluid analyses of amyloid-$β_{1-42}$.

Patients with Alzheimer's disease dementia were included if they met the DSM-5 criteria for dementia and showed abnormal amyloid-β accumulation on $^{18}$F-Flutemetamol PET or cerebrospinal fluid analyses of amyloid-$β_{1-42}$.

All subjects gave written informed consent to participate in this study and underwent a neuropsychological battery of tests that assessed global cognitive status (mini-mental state examination) (*Folstein et al., 1975*), episodic memory (delayed word list recall from the ADAS-Cog: Alzheimer's Disease Assessment Scale – Cognitive Subscale) (*Rosen et al., 1984*), attention functions (Trail Making Test A), and visuospatial abilities (clock-drawing test) (*Manos and Wu, 1994*). For the episodic memory and attention tests, higher scores indicated worse cognitive performance. A few subjects did not complete some of the previous cognitive tests (one subject had missing MMSE scores, six had missing delayed word recall scores, 33 had missing clock-drawing test scores, and 10 had missing Trail Making Test A scores).

This study received ethical approval from the Regional Ethical Review Board of Lund University (Dnr 2008–695, 2008–290, 2010–156), the Swedish Medicines and Products Agency (Dnr 151:2012/4552, 5.1-2014-62949), and the Radiation Safety Committee of Skåne University Hospital in Sweden.

### Image acquisition

In this multimodal neuroimaging study, all 117 subjects underwent $^{18}$F-Flutemetamol PET on a Philips Gemini TF 16 scanner, $^{18}$F-Flortaucipir PET on a General Electrics Discovery 690 scanner and structural MRI on a Siemens Tim Trio 3T scanner. In addition, 101 subjects also underwent resting-state functional MRI (of which 13 were excluded) and 88 subjects underwent diffusion tensor imaging (of which four were excluded) using a Siemens Tim Trio 3T scanner (*Figure 1*).

$^{18}$F-Flutemetamol PET images were acquired 90 to 110 min after injection of 185 MBq $^{18}$F-Flutemetamol and reconstructed into 4 × 5 frames using the line-of-response row-action maximum-likelihood algorithm (*Palmqvist et al., 2014*).

$^{18}$F-Flortaucipir PET images were acquired 80 to 100 min after injection of 370 MBq $^{18}$F-Flortaucipir, reconstructed into 4 × 5 frames using an iterative Vue Point HD algorithm with six subsets, 18 iterations with 3 mm filter and no time-of-flight correction (*Hahn et al., 2017*).

The structural T1-weighted images were acquired using a magnetization-prepared rapid gradient echo sequence with 176 slices, repetition time: 1950 ms, echo time: 3.4 ms, inversion time: 900 ms, flip angle: 9°, and 1 mm isotropic voxels.

Resting-state functional MRI images were acquired using a gradient-echo planar imaging pulse sequence with the following parameters: 180 volumes, 33 slices, repetition time: 2000 ms, echo time: 30 ms, 3 mm isotropic voxels (*Palmqvist et al., 2017*).

Finally, diffusion tensor imaging scans were acquired with 64 diffusion-weighted directions at a b value of 1000 s/mm2 using an echo-planar imaging sequence with 65 axial slices, repetition time: 8200 ms, echo time: 86 ms, no inter-slice gap, and 2 mm isotropic voxels.

## PET image preprocessing

The $^{18}$F-Flutemetamol and $^{18}$F-Flortaucipir PET images were motion-corrected, time-averaged and coregistered to their skull stripped T1-weighted images. These images were further normalized by a reference region (whole cerebellum, brain stem and eroded subcortical white matter for $^{18}$F-Flutemetamol; inferior cerebellar gray matter for $^{18}$F-Flortaucipir) (*Landau et al., 2015*; *Maass et al., 2017*), warped to MNI152 space, and smoothed using a 12 mm Gaussian filter. Warping to MNI was performed using the transformation parameters derived from warping the T1-weighted images to the MNI152 template of the FMRIB Software Library (FSL) software (version 5.0; https://fsl.fmrib.ox. ac.uk/fsl/fslwiki).

## Amyloid-β and tau PET joint independent component analysis

After image preprocessing, all $^{18}$F-Flutemetamol and $^{18}$F-Flortaucipir images were submitted to the joint independent component analysis using the Fusion ICA toolbox (*Calhoun et al., 2006*). This data driven analysis determines the hidden sources of a joint data distribution when the mixing matrices of different imaging modalities are stacked together. Thus, we used this approach to identify spatial patterns or components that are linked across $^{18}$F-Flutemetamol and $^{18}$F-Flortaucipir data. After identifying the spatial patterns, we thresholded them with a z score >2.0, which corresponds to a two-tailed significance value of p<0.05. These patterns were considered meaningful if their major clusters were not located in the white matter or in regions considered to represent off-target binding such as the basal ganglia for $^{18}$F-Flortaucipir data. Moreover, they were binarized and multiplied by a gray matter mask to further exclude any potential white matter or cerebrospinal fluid voxels from the analyses. Finally, for all individuals we extracted the mean standardized uptake value ratios (SUVR) of each amyloid-β and tau component.

In a secondary analysis, we also performed joint independent component analyses in the separate groups of subjects: amyloid-β positive controls, patients with mild cognitive impairment or patients with Alzheimer's disease. The aim of this analysis was to assess whether the networks of amyloid-β and tau accumulation would look entirely different in early, mild and advanced stages of Alzheimer's disease. The results of these analyses can be found in *Figure 2—figure supplement 2* and *Figure 2—figure supplement 3*. Most of the networks identified in the whole sample were also present in these separate groups, supporting our initial approach of performing the joint independent component analyses in the entire sample. In addition, despite showing similar spatial networks to the whole sample, the networks identified in the separate groups showed a patchier pattern and were noisier since they included more voxels from the white matter and cerebrospinal fluid. This suggests that our initial approach benefited from having the larger number of subjects that were included and produced less noisy components.

## T1-weighted image preprocessing and analysis

T1-weighted images were preprocessed using the Statistical Parametric Mapping (SPM) software (version 12; https://www.fil.ion.ucl.ac.uk/spm/). All images were segmented into gray matter, white

matter and cerebrospinal fluid. The gray and white matter images of each subject were used to create a study-specific template using the DARTEL toolbox (*Ashburner, 2007*). The gray matter images were normalized to this template, while preserving the total amount of gray matter volume and smoothed using a 12 mm Gaussian filter. The total intracranial volume was also calculated in these analyses as the sum of the gray matter, white matter and cerebrospinal fluid volumes. This variable was included in the statistical analyses to adjust for potential differences in head size.

To assess the relationship between gray matter volume and the SUVRs of the amyloid-β and tau networks, we used a gray matter mask containing all the regions that show atrophy in Alzheimer's disease identified in a recent meta-analysis (*Darby et al., 2019*). This mask was generously provided to us by the authors and included the precuneus, posterior cingulate, angular gyri, paracingulate gyrus, middle temporal gyri, medial frontal gyrus and hippocampus (*Darby et al., 2019*). We normalized this mask, binarized it and coregistered it to the preprocessed gray matter images of our sample and extracted the gray matter volumes within this mask from each subject. These volumes were included in correlation analyses to assess whether gray matter regions that are vulnerable to Alzheimer's disease are associated with the amyloid-β and tau patterns identified in the current study.

## Functional MRI preprocessing and analysis

Resting-state functional MRI images were preprocessed with SPM using the following steps: removal of the first five volumes, realignment, slice-timing correction, coregistration and normalization to the T1-weighted images, smoothing with a 8 mm Gaussian filter, and band-pass filtering). Thirteen subjects were excluded due to errors during image normalization or due to excessive motion (>3 mm) so the final sample size with functional MRI was 88 subjects.

To assess the relationship between the PET networks and the activation signals from our functional MRI images, we downloaded the resting-state networks identified by *Biswal et al. (2010)* in 1414 volunteers from the 1000 Functional Connectomes Project, which are available at http://www.nitrc.org/projects/fcon_1000/. These networks included the visual (primary, secondary, extra-striate), default-mode (anterior, posterior), fronto-parietal (left, right), anterior cingulate, limbic, dorsal attention, salience (anterior, posterior), executive control (lateral, medial), basal ganglia, language, somatosensory and sensorimotor networks. We excluded the white matter and cerebellar networks provided by *Biswal et al. (2010)* because [18]F-Flutemetamol and [18]F-Flortaucipir uptake in the white matter is thought to represent off-target binding and because the cerebellum was used as a reference region to normalize [18]F-Flutemetamol and [18]F-Flortaucipir images.

We normalized our functional MRI scans to the same template used by *Biswal et al. (2010)* for spatial image normalization. Then, we binarized resting-state network maps and coregistered them to the preprocessed functional MRI images of our sample. Finally, we extracted the mean time-series of each map from all subjects and included them in correlation analyses to assess whether they were associated with the mean SUVR values of amyloid-β and tau within networks identified in the joint independent component analysis.

## Diffusion tensor imaging preprocessing and analysis

Diffusion tensor images were corrected for distortions caused by eddy currents and head motion, and skull-striped using FSL. A diffusion tensor model was then fitted at each voxel to calculate the fractional anisotropy and mean diffusivity maps for each subject using the Diffusion Toolbox (*Behrens et al., 2007*).

To assess the relationship between the PET networks and white matter integrity, we downloaded the JHU white-matter tractography atlas (*Hua et al., 2008*). This atlas contains 11 white matter tracts, which were binarized and coregistered to the native fractional anisotropy and mean diffusivity images of our sample. These white matter tracts consisted of the hippocampal cingulum tract, the cingulate cingulum tract, the anterior thalamic radiation, the corticospinal tract, the inferior occipito-frontal fasciculus, the inferior longitudinal fasciculus, the superior longitudinal fasciculus, the superior longitudinal fasciculus temporal part, the uncinate fasciculus, the forceps major and forceps minor. Finally, we extracted the fractional anisotropy and mean diffusivity values of all white matter tracts from each subject.

There were four outliers in white matter integrity measures, which were excluded from the analyses.

## Statistical analyses

Differences between groups in clinical variables were analyzed using Chi-squared tests for binary variables and Mann-Whitney tests for ordinal or continuous variables.

To assess the value of different amyloid-β and tau PET networks in discriminating different groups, we compared the SUVRs of these networks between amyloid-β negative and positive controls in addition to patients with mild cognitive impairment and patients with Alzheimer's disease. These analyses were carried out using permutation tests (10,000 replicates), while controlling for age and sex. Adjustment for multiple comparisons was performed using false discovery rate (FDR) corrections (*Benjamini and Hochberg, 1995*) at q < 0.05.

## Spatial overlap with the functional MRI networks

To determine the overlap between the spatial patterns of amyloid-β and tau with the well-known functional MRI networks provided by *Biswal et al. (2010)* we computed the Dice similarity coefficient for each pair of networks (PET, functional MRI) after they had been binarized.

This coefficient was calculated as follows: *2nj/(nx+ny)*, where *nj* is the volume intersection between the image volume *nx* and *ny*, which is divided by the sum of the respective image volumes *nx* and *ny*.

The Dice similarity coefficient can be interpreted in the following way: poor overlap (<0.2), fair overlap (0.2–0.4), moderate overlap (0.4–0.6), good overlap (0.6–0.8), and excellent overlap (>0.8) (*Hoenig et al., 2018*).

## Association between the PET networks with brain atrophy, functional networks, white matter tracts and cognition

To assess the relationship between the PET networks SUVRs with brain atrophy, brain connectivity and cognition, we carried out partial correlation analyses in the separate groups using Spearman's Rho. The variables included in these analyses were the mean gray matter volumes of a composite mask containing vulnerable regions to Alzheimer's disease (*Darby et al., 2019*), the mean time series of the resting-state networks (functional connectivity), the fractional anisotropy and mean diffusivity values of white matter tracts (anatomical connectivity), or the scores in global, memory, attention and visuospatial tests (cognition) of our sample. Age, sex and presence of cognitive impairment (cognitively normal or mild cognitive impairment/dementia) were included as covariates in all analyses. Moreover, intracranial volume was included as an additional covariate in the gray matter volume analyses, whereas education was included as an additional covariate in the correlation analyses that included cognitive measures. To correct for multiple comparisons, FDR corrections (q < 0.05) were applied.

Since the PET networks were identified in a larger sample than the subsample that had functional MRI and diffusion tensor imaging data, we also performed a secondary analysis to identify the amyloid-β and tau networks in the subsample of our study that had all imaging modalities (n = 67): [18]F-Flutemetamol, [18]F-Flortaucipir, T1-weighted, functional MRI and diffusion tensor imaging. These analyses included 10 amyloid-β negative cognitively normal subjects, 24 amyloid-β positive cognitively normal subjects, 14 patients with mild cognitive impairment and 19 patients with Alzheimer's disease. The results of this analysis can be found in *Figure 2—figure supplement 4*. Most of the networks present in the whole sample could also be identified in this subsample with all imaging modalities. For this reason we decided to use the PET networks identified in the whole cohort in all analyses since they included a larger number of subjects and were therefore potentially more representative of different stages of Alzheimer's disease as well as more stable and robust.

## Acknowledgements

Work at the authors' research center was supported by the European Research Council (OH), the Swedish Research Council (JBP, EW, OH), the Knut and Alice Wallenberg foundation (OH), the Marianne and Marcus Wallenberg foundation (OH), the Strategic Research Area MultiPark (Multidisciplinary Research in Parkinson's disease) at Lund University (OH, SP), the Swedish Alzheimer

Foundation (JBP, RO, RS, SP, EW, OH), the Swedish Brain Foundation (JBP, OH), The Parkinson foundation of Sweden (OH), The Parkinson Research Foundation (OH), the Skåne University Hospital Foundation (OH), the Swedish federal government under the ALF agreement (OH), the Swedish Foundation for Strategic Research (SSF) (EW), the Strategic Research Programme in Neuroscience at Karolinska Institutet (StratNeuro) (JBP, EW), Stiftelsen Olle Engkvist Byggmästare (EW), Birgitta och Sten Westerberg (EW), and the Åke Wiberg Foundation (EW). Doses of $^{18}$F-Flutemetamol injections were sponsored by GE Healthcare. The precursor of $^{18}$F-Flortaucipir was provided by AVID radiopharmaceuticals.

## Additional information

### Competing interests

Oskar Hansson: has acquired research support (for the institution) from Roche, GE Healthcare, Biogen, AVID Radiopharmaceuticals, Fujirebio, and Euroimmun. In the past 2 years, he has received consultancy/speaker fees (paid to the institution) from Biogen, Roche, and Fujirebio. The other authors declare that no competing interests exist.

### Funding

| Funder | Grant reference number | Author |
| --- | --- | --- |
| European Research Council | | Oskar Hansson |
| Alzheimerfonden | | Joana B Pereira<br>Rik Ossenkoppele<br>Sebastian Palmqvist<br>Ruben Smith<br>Eric Westman<br>Oskar Hansson |
| Swedish Research Council | | Joana B Pereira<br>Eric Westman<br>Oskar Hansson |
| Knut and Alice Wallenberg Foundation | | Oskar Hansson |
| Marianne and Marcus Wallenberg Foundation | | Oskar Hansson |
| Swedish Brain Research | | Joana B Pereira |
| Lund University | Strategic Research Area MultiPark (Multidisciplinary Research in Parkinson's disease) | Oskar Hansson<br>Sebastian Palmqvist |
| Swedish Brain Foundation | | Joana B Pereira<br>Oskar Hansson |
| Parkinsonfonden | | Oskar Hansson |
| Parkinson Research Foundation | | Oskar Hansson |
| Skåne University Hospital | | Oskar Hansson |
| Swedish federal government under the ALF agreement | | Oskar Hansson |
| Swedish Foundation for Strategic Research | | Eric Westman |
| Karolinska Institutet | Strategic Research Programme in Neuroscience | Joana B Pereira<br>Eric Westman |
| Stiftelsen Olle Engkvist Byggmästare | | Eric Westman |
| Birgitta och Sten Westerberg | | Eric Westman |

| Åke Wiberg Foundation | Eric Westman |
| --- | --- |

The funders had no role in study design, data collection and interpretation, or the decision to submit the work for publication.

## Author contributions

Joana B Pereira, Conceptualization, Formal analysis, Investigation, Visualization, Methodology; Rik Ossenkoppele, Sebastian Palmqvist, Ruben Smith, Eric Westman, Writing - review and editing; Tor Olof Strandberg, Data curation; Oskar Hansson, Conceptualization, Resources, Supervision, Funding acquisition, Project administration

## Author ORCIDs

Joana B Pereira (iD) https://orcid.org/0000-0002-4604-2711

## Ethics

Human subjects: This study received ethical approval from the Regional Ethical Review Board of Lund University (Dnr 2008-695, 2008-290, 2010-156), the Swedish Medicines and Products Agency (Dnr 151:2012/4552, 5.1-2014-62949), and the Radiation Safety Committee of Skåne University Hospital in Sweden. All participants provided informed consent before being included in the study.

## Decision letter and Author response

Decision letter https://doi.org/10.7554/eLife.50830.sa1
Author response https://doi.org/10.7554/eLife.50830.sa2

## Additional files

### Supplementary files

• Supplementary file 1. Association between functional MRI network signals with the amyloid-β and tau networks SUVRs.

• Supplementary file 2. Association between gray matter volumes in vulnerable regions to Alzheimer's disease with the amyloid-β and tau networks SUVRs.

• Transparent reporting form

### Data availability

Source data files have been provided for Figures 4 and 5. The source data for the rest of the analyses performed in this study can be requested from Prof. Oskar Hansson (Oskar.Hansson@med.lu.se), after signing a material transfer agreement from Lund University that ensures that the data will only be used for the sole purpose of replicating procedures and results presented in the article and as long as data transfer is in agreement with EU legislation on the general data protection regulation and decisions by the Ethical Review Board of Sweden and Region Skåne.

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
