## [Decision Letter]

**Acceptance summary:**

This is an interesting and timely study that tests the different spatial affinities of tau and amyloid on brain network integrity in aging and Alzheimer's disease. A clear strength of the study is the use of multimodal neuroimaging techniques (resting-state functional connectivity, diffusion weighted imaging, structural imaging) within the same cohort of individuals to explore the effects of amyloid versus tau aggregation, and linking these metrics to cognition. The study provides one of the most comprehensive examinations of the interrelation and overlap between amyloid and tau networks to date, and offers a solid foundation for future research exploring the interaction between these two proteins in the pathogenesis of Alzheimer's disease.

**Decision letter after peer review:**

Thank you for submitting your article "Amyloid and tau accumulate across distinct spatial networks and are differentially associated with brain connectivity" for consideration by *eLife*. Your article has been reviewed by two peer reviewers, one of whom is a member of our Board of Reviewing Editors, and the evaluation has been overseen by Timothy Behrens as the Senior Editor. The reviewers have opted to remain anonymous.

The reviewers have discussed the reviews with one another and the Reviewing Editor has drafted this decision to help you prepare a revised submission.

This is an interesting and timely study that tests the different spatial affinities of tau and amyloid on brain network integrity in aging and Alzheimer's disease. A clear strength of the study is the use of multimodal neuroimaging techniques (resting-state functional connectivity, diffusion weighted imaging, structural imaging) within the same cohort of individuals to explore the effects of amyloid versus tau aggregation, and linking these metrics to cognition. The study thus provides one of the most comprehensive examinations of the interrelation and overlap between amyloid and tau networks to date.

Major points:

1) While the authors have done an impressive job in recruiting relatively large numbers for each of their groups of interest, participants were excluded or removed for various reasons along the way, yet a final breakdown of the participant numbers in each separate group is not provided. This is not a trivial point as it is not clear whether a systematic bias is present in terms of the participant numbers in the final dataset. Moreover, is the subset of individuals who completed all imaging modalities representative of their respective cohort in terms of cognitive profile and amyloid/tau aggregation? These data should be clearly presented, possibly as a flow-chart, identifying the number of cases that were excluded at each stage (and from which group), leading through to the final numbers for each of the 4 groups. This information should also be presented in the figure legends and footnotes for tables so the reader can see how many cases are included in each set of analyses, and whether they are representative of their group in general.

2) Throughout the paper, the authors claim to have assessed the spatial networks associated with amyloid and tau accumulation "across the entire Alzheimer's disease spectrum", however this statement is misleading as it implies that patients at mild, moderate to severe stages of AD (perhaps graded using a clinical staging tool such as the CDR) have been included in the study, which is not the case. Further, all of the analyses are correlational, yet are interpreted in a somewhat causal manner. From these results, as currently presented, we cannot draw any definitive conclusions regarding the spread of amyloid-β and tau across the different brain networks to influence gray matter atrophy and cognitive decline with disease evolution. In the absence of diffusion models or graph-based network analysis, it is important to temper these claims of progression/spread throughout the manuscript.

3) The authors performed a joint ICA analysis on the amyloid-β and the tau data derived from all of subjects (including controls, MCI and AD). This analysis could be problematic, since different groups of subjects could show different components or spatial distribution in the amyloid-β and the tau. Is it possible to perform this ICA for each group separately? The authors mentioned the optimal number of components (i.e., 11) identified in the joint independent component analysis of 18F-Flutemetamol and 18F-Flortaucipir data but it is not clear how they determined the optimal number. The authors mapped the fMRI components from Biswal et al., 2010, onto the amyloid-β and the tau components. Given that the fMRI components (Biswal et al., 2010) were derived from young adults, the authors need to perform image transformations if they used this information in the older populations.

4) The authors performed many correlation analyses among variables (e.g., brain and cognition) across all of the participants (Figure 3 and 4). Given these variables showed different values among groups, it is not surprising to find these correlations. The authors need to perform these analyses within each group.

5) The Dice coefficients are helpful to appreciate the extent of spatial overlap between the amyloid and tau networks with existing resting-state functional networks. However, the Dice coefficients observed here were rather weak (the highest coefficient is.48, which is moderate at best), and do not demonstrate that amyloid/tau have an affinity for one network over another, i.e., the information presented here doesn't allow us to dissociate between the two pathologies. As such, the authors should temper their discussion of these findings.

---

## [Author Response]

This is an interesting and timely study that tests the different spatial affinities of tau and amyloid on brain network integrity in aging and Alzheimer's disease. A clear strength of the study is the use of multimodal neuroimaging techniques (resting-state functional connectivity, diffusion weighted imaging, structural imaging) within the same cohort of individuals to explore the effects of amyloid versus tau aggregation, and linking these metrics to cognition. The study thus provides one of the most comprehensive examinations of the interrelation and overlap between amyloid and tau networks to date.Major points:1) While the authors have done an impressive job in recruiting relatively large numbers for each of their groups of interest, participants were excluded or removed for various reasons along the way, yet a final breakdown of the participant numbers in each separate group is not provided. This is not a trivial point as it is not clear whether a systematic bias is present in terms of the participant numbers in the final dataset. Moreover, is the subset of individuals who completed all imaging modalities representative of their respective cohort in terms of cognitive profile and amyloid/tau aggregation? These data should be clearly presented, possibly as a flow-chart, identifying the number of cases that were excluded at each stage (and from which group), leading through to the final numbers for each of the 4 groups. This information should also be presented in the figure legends and footnotes for tables so the reader can see how many cases are included in each set of analyses, and whether they are representative of their group in general.

We understand the reviewer’s concerns and have included a new figure in the manuscript (Figure 1), which is a flow-chart containing this information. In this flow-chart we included the total number of subjects with: a) 18F-Flutemetamol PET, 18F-Flortaucipir PET, T1-weighted MRI; b) resting-state functional MRI; and C) diffusion tensor imaging. For each of these 3 samples we stated how many subjects belonged to each group and the number of excluded subjects. In addition, since the final analyses included in the study were based on the previous 3 samples and not the subsample with all five imaging modalities (n = 67), we included the information regarding this smaller subsample in the text and supplementary material to avoid any potential misunderstandings regarding which sample was included in our final analyses:

“Since the PET networks were identified in a larger sample than the subsample that had functional MRI and diffusion tensor imaging data, we also performed a secondary analysis to identify the amyloid-β and tau networks in the subsample of our study that had all imaging modalities (n = 67): 18F-Flutemetamol, 18F-Flortaucipir, T1-weighted, functional MRI and diffusion tensor imaging. […] For this reason we decided to use the PET networks identified in the whole cohort in all analyses since they included a larger number of subjects and were therefore potentially more representative of different stages of Alzheimer’s disease as well as more stable and robust.”

Finally, we have also stated the number of subjects included in each figure legend and table footnotes, except Table 1 because the numbers are included within the table.

2) Throughout the paper, the authors claim to have assessed the spatial networks associated with amyloid and tau accumulation "across the entire Alzheimer's disease spectrum", however this statement is misleading as it implies that patients at mild, moderate to severe stages of AD (perhaps graded using a clinical staging tool such as the CDR) have been included in the study, which is not the case. Further, all of the analyses are correlational, yet are interpreted in a somewhat causal manner. From these results, as currently presented, we cannot draw any definitive conclusions regarding the spread of amyloid-β and tau across the different brain networks to influence gray matter atrophy and cognitive decline with disease evolution. In the absence of diffusion models or graph-based network analysis, it is important to temper these claims of progression/spread throughout the manuscript.

We fully agree with the reviewer and have attenuated all claims related to the spread of amyloid and tau pathology in the manuscript. In addition, we have included the limitations related to the correlation analyses in the Discussion:

“We would like to highlight that a large part of the analyses carried out in the current study were based on multiple correlation analyses in all amyloid-β positive individuals. […] Thus, our analyses do not allow drawing definitive conclusions regarding the spread of amyloid-β and tau across different brain networks.”

3) The authors performed a joint ICA analysis on the amyloid-β and the tau data derived from all of subjects (including controls, MCI and AD). This analysis could be problematic, since different groups of subjects could show different components or spatial distribution in the amyloid-β and the tau. Is it possible to perform this ICA for each group separately? The authors mentioned the optimal number of components (i.e., 11) identified in the joint independent component analysis of 18F-Flutemetamol and 18F-Flortaucipir data but it is not clear how they determined the optimal number. The authors mapped the fMRI components from Biswal et al., 2010, onto the amyloid-β and the tau components. Given that the fMRI components (Biswal et al., 2010) were derived from young adults, the authors need to perform image transformations if they used this information in the older populations.

We had already performed the joint ICA analyses in the separate groups suggested by the reviewer, which were included in Figure 2—figure supplement 2 and Figure 2—figure supplement 3. These analyses showed that the spatial patterns of amyloid and tau in the different groups were similar to the ones observed in the entire sample. However, the patterns identified in each group were noisier since they included some white matter and cerebrospinal fluid voxels. Moreover, these patterns were also patchier, probably due to the lower number of subjects in each group. In addition, since one of the aims of our study was to compare the amyloid and tau component SUVRs between groups, we needed the components to be the same across all participants for these comparisons to be meaningful. For all these reasons we decided to keep the components of the joint ICA in the entire sample as our main analyses. However, we mention these analyses in the separate groups in the Materials and methods section and show them results in Figure 2—figure supplement 2 and Figure 2—figure supplement 3 so that this information is available to the readers:

“In a secondary analysis, we also performed joint independent component analyses in the separate groups of subjects: amyloid-β positive controls, patients with mild cognitive impairment or patients with Alzheimer’s disease. […] This suggests that our initial approach benefited from having the larger number of subjects that were included and produced less noisy components.”

Regarding the method used to identify the optimal number of components in our sample, we have included this information in the Results section of our manuscript:

“The optimal number of components identified in the joint independent component analysis of 18F-Flutemetamol and 18F-Flortaucipir data was eleven using the minimum description length criterion (Li et al., 2007). This criterion selects the best hypothesis (a model and its parameters) for a given set of data as the one that leads to the best compression of the data. This method is provided within the Fusion ICA toolbox (Calhoun et al., 2006), which we used to perform the joint independent component analysis.”

Finally, regarding the fMRI components, we agree with the reviewer that image transformations are needed to be able to use the functional networks provided by Biswal et al., 2010, in our sample. We did perform these transformations but we did not describe this clearly in the manuscript. We have now made this clearer in the new version of the manuscript:

“We normalized our functional MRI scans to the same template used by Biswal et al., 2010, for spatial image normalization. Then, we binarized Biswal’s (2010) resting-state network maps and coregistered them to the preprocessed functional MRI images of our sample. Finally, we extracted the mean time-series of each map from all subjects and included them in correlation analyses to assess whether they were associated with the mean SUVR values of amyloid-β and tau within networks identified in the joint independent component analysis.”

4) The authors performed many correlation analyses among variables (e.g., brain and cognition) across all of the participants (Figure 3 and 4). Given these variables showed different values among groups, it is not surprising to find these correlations. The authors need to perform these analyses within each group.

Following the reviewer’s suggestions we performed the correlation analyses in the separate groups. These analyses showed several correlations between the PET networks, cognition and MRI measures. However, we would prefer keeping the correlation analyses across all amyloid positive individuals as the main analysis of our manuscript. The aim of our study was to assess the relationship between different markers across the AD continuum, in line with the recent criteria proposed by the NIA-AA that AD should be regarded as a continuum and not as distinct clinically defined entities (cognitively normal, mild cognitive impairment, dementia) (Jack et al., 2018). In addition, the number of subjects in each group of our study was small, making these analyses not very reliable. Finally, the analyses in each group would increase the number of correlations by a factor of 3, increasing the chance of finding false positives. Although we agree with the reviewer that the variables we are correlating are different between the groups, we would like to highlight that the correlation plots (see for example Figure 4 and 5) do not show that the results are being driven by a single group but that the subjects seem to be well distributed across the regression line. In addition, these analyses were controlled by age, sex and presence of cognitive impairment in order to adjust for the confounding effects of these variables. Thus, for the sake of transparency, we have performed the requested analyses for the reviewers but we would prefer not changing our main analyses. If the reviewers still feel that is important to have these analyses in the manuscript, we would be happy to include them in supplementary material.

5) The Dice coefficients are helpful to appreciate the extent of spatial overlap between the amyloid and tau networks with existing resting-state functional networks. However, the Dice coefficients observed here were rather weak (the highest coefficient is.48, which is moderate at best), and do not demonstrate that amyloid/tau have an affinity for one network over another, i.e., the information presented here doesn't allow us to dissociate between the two pathologies. As such, the authors should temper their discussion of these findings.

We agree with the reviewer that the spatial overlap between the PET and the functional MRI networks were not strong in our study. To make sure that this is clear to the readers, we have now changed our description of these results across all sections of the manuscript to highlight that the overlap between these networks was fair (dice coefficient: 0.2-0.4) or moderate (dice coefficient: 0.4-0.6).